# Herb Polysaccharide-Based Drug Delivery System: Fabrication, Properties, and Applications for Immunotherapy

**DOI:** 10.3390/pharmaceutics14081703

**Published:** 2022-08-15

**Authors:** Yubiao Cao, Zhuowen Chen, Liangliang Sun, Yameng Lin, Ye Yang, Xiuming Cui, Chengxiao Wang

**Affiliations:** 1School of Life Science and Technology, Kunming University of Science and Technology, Kunming 650500, China; 2Key Laboratory of Sustainable Utilization of Panax Notoginseng Resources of Yunnan Province, Kunming 650500, China

**Keywords:** herb polysaccharides, drug delivery system, immune regulation, structure modification, application

## Abstract

Herb polysaccharides (HPS) have been studied extensively for their healthcare applications. Though the toxicity was not fully clarified, HPS were widely accepted for their biodegradability and biocompatibility. In addition, as carbohydrate polymers with a unique chemical composition, molecular weight, and functional group profile, HPS can be conjugated, cross-linked, and functionally modified. Thus, they are great candidates for the fabrication of drug delivery systems (DDS). HPS-based DDS (HPS-DDS) can bypass phagocytosis by the reticuloendothelial system, prevent the degradation of biomolecules, and increase the bioavailability of small molecules, thus exerting therapeutic effects. In this review, we focus on the application of HPS as components of immunoregulatory DDS. We summarize the principles governing the fabrication of HPS-DDS, including nanoparticles, micelles, liposomes, microemulsions, hydrogels, and microneedles. In addition, we discuss the role of HPS in DDS for immunotherapy. This comprehensive review provides valuable insights that could guide the design of effective HPS-DDS.

## 1. Introduction

Polysaccharides, which are natural high-molecular-weight compounds, are important biopolymers. They are composed of monosaccharide subunits joined by glycosidic linkages [1]. Typically, polysaccharides contain more than 10 monosaccharides, and their molecular weight (Mw) ranges from thousands to millions Da [2]. So far, more than 300 kinds of natural polysaccharides have been identified from fungi, plants, animals, algae, and bacteria [3]. Hence, they represent a fundamental renewable resource of biomass that can replace fossil-based polymeric materials in commodities and industrial applications [4]. Compared with synthetic materials, polysaccharides have unique biological properties, include hydrophilicity [5], good stability [6], abundant availability [7], bioadhesion [8], and low toxicity. Moreover, they often show low immunogenicity [9], biodegradability [10], and ease of chemical modification [11]. Polysaccharides remain popular candidates for fabricating biomaterials for drug delivery systems (DDS). For example, hyaluronic acid (HA), which can form gels, is used to produce scaffolds with very intriguing mechanical properties [12]. Chitosan (CS) [13], known for its anti-bacterial and film-forming properties, is widely used in advanced nanotechnology. Alginate is a natural polysaccharide that can easily be chemically modified or combined with other components to produce various functionalities. Furthermore, alginate derivatives are appealing not only because they can be used in biomedicine but also because of their potential in bioelectronics, as they can act as the interface between human tissues and electronic devices or even serve as electronic components themselves [14].

In addition to beneficial material properties, some polysaccharides also have significant biological activities [15]. In particular, polysaccharides derived from herbs, such as various Chinese herb medicines, have attracted attention for their immunostimulatory activities, although the detailed mechanism of their effects is currently under exploration [16]. Numerous studies indicate that HPS can be used in many fields and have diverse therapeutic properties and immunostimulatory activities [17], including antioxidation [18], antitumor [19], and wound-healing effects [20]. For example, *Panax ginseng* polysaccharides can regulate the immune system and induce anti-cancer effects via anti-complementary activity, peritoneal macrophage activation, and cytotoxicity against natural killer (NK) cells [21]. *Astragalus* is a tonic herb widely used to improve the immune status of the body. According to reports, polysaccharides from *Astragalus* can regulate the activities of immune organs and immune cells and promote the release of immune mediators (such as IL-2, IL-6, TNF-α, and IFN-γ) [22]. *Dendrobium officinale* polysaccharides can remarkably inhibit tumor growth by promoting macrophage polarization from the M2 phenotype to the M1 phenotype in the tumor microenvironment (TME) [23]. *Bletilla striata* polysaccharides are used to prepare composite aerogels that exhibit strong antibacterial and hemostatic activity, reducing inflammation and rapidly enhancing wound re-epithelialization by down-regulating TNF-α and up-regulating VEGF-A. These results prove that HPS possess excellent immunoregulatory bioactivities [24].

The combination of potential pharmacological activity and useful (bio)physical properties increases the value of HPS as building blocks for DDS, which are active treatment agents. Previous studies have reported the synthesis of HPS-DDS, such as nanoparticles (NPs) [25,26], micelles [27], liposomes [28], microemulsions [29], hydrogels [30], and microneedles [31], for the treatment of various diseases. These studies show that HPS can serve as drug carriers [32], stabilizers [33], solubilizers [34], and emulsifiers [35]. Considering their bioactivity, these HPS can also act as adjuvants [36], immunomodulators [37], and bioactive macromolecular drugs [38] that exert synergistic effects or amplify the therapeutic activities of primary drugs. 

Several reviews have focused on DDS based on natural polysaccharides such as CS [39], HA [40], dextran [41], and alginate [42] and highlighted their material properties. However, the use of HPS for DDS with immunomodulatory activity has rarely been reviewed. In addition, the application of HPS in DDS have rarely been reviewed. Accordingly, this review discusses the main essential properties of HPS-based DDS, including their chemical properties and immunostimulatory mechanisms. In addition, the principle of HPS-DDS fabrication and the role of HPS in DDS are summarized. Finally, this review discusses the recent progress in the applications of HPS-DDS and addresses the prospects for future studies. The aim of this review is to offer a comprehensive understanding of the current research and development of HPS-DDS and provide a framework for the fabrication of novel HPS-DDS in the future. 

## 2. Immunomodulatory Activity of HPS

More than 1000 kinds of HPS from herb plants have been demonstrated to have immune-related activities [43,44]. Table 1 shows the effect of various HPS (e.g., *Bletilla striata, Angelica sinensis,* and *Astragalus membranaceus*) on immune-related diseases. Notably, polysaccharides with immunomodulatory activities can directly or indirectly interact with the immune system [45]. In general, HPS exert immune effects by regulating immune cells (macrophages, dendritic cells (DCs), T cells, NK cells, etc.) [46] and promoting cell–cell communication [47]. Once a pathogen enters the immune system, it is recognized and processed by antigen-presenting cells (APC). These cells present antigens to T cells and induce T cell differentiation, leading to a series of immune responses [48]. For example, after microneedle administration, DCs prime T cells, thereby playing an immune role [49]. In TME, macrophages activate T cells, and T cells can also be affected directly [50,51]. According to the literature, traditional HPS such as honeysuckle polysaccharides [52], tea polysaccharides [53], and lychee polysaccharides [54] can promote the development of immune organs, participate in the body’s immune response and defense, and thus maintain an immune balance.

HPS perform immunomodulation through multiple modes and targets in order to regulate immune function, thereby improving biological immunity at the organ, tissue, cellular, and molecular levels [55,56]. Studies have shown that most HPS perform immunomodulation by binding to cell surface receptors, stimulating linked signaling channels, and promoting cytokine expression [45]. At present, the key HPS receptors identified include TLR-4 [57], scavenger receptor [58], mannose receptors (MRs) [59], MAPK [60], CR3/CD14 [61], and Dectin-1 [62]. In addition, other receptors and signaling channels can also mediate the immunomodulatory effects of HPS [45]. One example of the typical receptor-mediated immunomodulatory mechanisms underlying the effects of HPS includes the action of *Angelica sinensis* polysaccharides (ASPs) on renal leukocytes. These polysaccharides act on TLR-2 and its downstream MAPK and NF-κB signaling pathways as well as on molecules downstream of TLR4, such as MAPK and NF-κB. Accordingly, they upregulate MHC-II, TNF-α, and IL-8 [63]. *Astragalus* polysaccharides (APSs) act on dendritic cells (DCs) via the JAK/STAT signaling pathway [64]. *Hibiscus sabdariffa Linn.* polysaccharides act on RAW264.7 cells by activating the MAPK and NF-κB signal pathway, phosphorylating ERK, JNK, p38, and p65, and promoting the expression of iNOS, IL-6, IL-1α, TNF-α, and IL-6 [65]. Together, the findings suggest that the immunomodulatory effects of HPS do not follow a single pattern. Instead, these polysaccharides regulate the body’s immune function via multiple pathways and targets, thereby boosting biological immunity. It is worth noting that the immunomodulatory effects of HPS on different immune cells involve different receptors and signaling pathways, and these polysaccharides also exert regulatory effects on intercellular messenger molecules.

Numerous studies have shown that molecular weight [66], monosaccharide composition and ratio [67], glycoside bond type [68], and grading structure [69] can influence the immune activity of HPS. An excessively high molecular weight can reduce the exposure of the active sites on HPS, thus limiting their bioactivity. However, if the molecular weight is too low, the number of active functional groups becomes insufficient, leading to poor bioactivity [70,71,72]. The monosaccharide composition also plays an important role in the immune activity of HPS. Glucose, galactose, and mannose are more easily recognized by receptors on the cell surface and trigger a series of immune responses via macrophages [45]. Glucuronic acid can enhance the immune activity of polysaccharides [73]. The β-(1→3) glycosidic bond is a non-reducing glycosidic bond, increasing the stability and tolerance of polysaccharides and reducing its effect on polysaccharides activity [74,75,76]. 

The charge and the abundant active groups on the surface of HPS can also affect its immune activity. Research shows that the negatively charged HPS can avoid the attack of macrophages by electrostatic attraction, which can improve the efficacy of the immunomodulatory activity [77]. That is because a large number of negatively charged polyanionic HPS are more likely to interact with basic proteins in the body, recognize and control the activity of immune cells, reduce immunogenic response, and thus regulate their immune activity [78]. In addition, Wanwan Xiao et al. [79] showed that the four types of *Platycodon grandiflorum* Polysaccharides show different immune activity, and the higher uronic acid content exhibited better immune-stimulating activity. Due to the carboxyl groups being abundant in acidic polysaccharides, it has the ability to provide hydrogen to pathogenic sites for reducing inflammation [80]. Moreover, the expression of uronic acid receptor on of tumor tissues and immune cells is higher, which is conducive to the accumulation of HPS [81].

**Table 1 pharmaceutics-14-01703-t001:** Summary of herb polysaccharides and their bioactivities.

Herb Polysaccharides	Molecular Weight	Monosaccharide Composition and Proportion of Monosaccharide	Main Types of Glycosidic Bonds	Pharmacological Effect	References
Bletilla striata polysaccharides	8.354 × 10^4^ Da 1.26 × 10^4^ Da	Man and Glc in the ratio of 4.0:1.0 Man and Glc in the ratio of 3.0:1.0	Backbone composed of β-1,4-linked Manp Backbone composed of β-1,4-linked Glcp	Immunomodulatory	[76]
	1.46 × 10^5^ Da	Man and Glc in the ratio of 2.4:1.0	α-Man and β-Glc residues	Anti-inflammatory	[82]
	Mw:3.73 × 10^5^ g/mol Mn:6.75 × 10^4^ g/mol	Man and Glc in the ratio of 2.946:1	Composed of 1,4-linked Glcp Branches composed of 1,3-linked Manp and 1,3-linked Glcp	Anti-inflammatory	[83]
	2.35 × 10^5^ Da	Man, Glc, and Gal in the ratio of 9:2.6:1.0	Composed of 1,4-linked Manp and 1,4-linked Glcp in a molar ratio of 2:1	Immunomodulatory	[84]
Angelica sinensis polysaccharides	1.0 × 10^5^ Da	Glc	α-(1→6)-glucan	Immunomodulatory	[85]
	44,000 Da	arabinose and galactose, as well as xylose, rhamnose and mannose	1,5 linked, galactose containing 1,4 and 1,4,6 linked units, xylose 1,4 linked, rhamnose 1,2 linked, and terminal mannose		[86]
American Ginseng polysaccharides		Arabinose, rhamnose, mannose, glucose, and glucose acid		Immunomodulatory	[87]
	3.1 kDa	Glucose (Glc) and galactose (Gal) in a molar ratio of 1:1.15		Anti-inflammatory	[88]
Astragalus membranaceus polysaccharides		Glucose, in addition to rhamnose, galactose, arabinose, xylose, mannose, glucuronic acid, and galacturonic acid		Immunoregulatory, antitumor, anti-inflammatory, and antiviral	[89]
Lycium barbarum L. polysaccharides	4.2 × 10^4^ Da 4.1 × 10^4^ Da	Glucose, mannose, and galactose Glc:Man:Gal = 1:0.4:0.1	(1→3)-linked α-d-glucopyranosyl residues (1→2)-linked α-d-mannopyranosyl residues a-d-(1,3)-Glcp a-d-(1,2)-Manp-p-d-Galp (1→2)-linked α-d-mannopyranosyl residues, β-D-galactopyranosyl residues	Anti-inflammatory	[90]
Yam polysaccharides	4.2 × 10^4^ Da	Glc:Man:Gal = 1:0.37:0.11	(1→3)-α-Glucopyranose as the main chain and β-galactopyranose-[(1→2)-α-mannopyranose]_3_-(1→2)-α-mannopyranose-(1→6) as the side chain	Immunomodulatory	[91]

## 3. General Rules of HPS-DDS

### 3.1. Preparation of HPS-DDS

In some cases, HPS can be transformed into nanoparticles through nanoprecipitation or nanosuspension [92]. However, HPS often need to be combined with other polymeric materials and drugs to prepare DDS. The strategies for combining HPS with polymers/drugs mainly include physical modification [93], chemical modification [94], electrostatic adsorption [56], and covalent linking [95] (Table 2).

#### 3.1.1. Self-Aggregation

When HPS serve as therapeutic agents, they can directly be transformed into NPs via self-aggregation without using additional copolymers [96]. Microfluidic technology appears to be a promising strategy for the preparation of HPS NPs due to its well-controlled, reproducible, and high-throughput nature [97,98]. Microfluidic systems consist of two infusion syringe pumps and a micro T-junction, which are linked by a capillary. To prepare NPs, HPS are dissolved in an aqueous solution to obtain a separate phase, while an anti-solvent, such as acetone, serves as the continuous phase. The two phases come together at the Micro-T junction owing to the action of the infusion pump. Accordingly, the polysaccharides experience a change in solvent at the micro scale. The polysaccharide unimers transform into nuclei under the high shear force between the two phases. These nuclei grow in size through a diffusion-limited process, and resulting in the formation of “kinetically locked” NPs. By adjusting the flow rate and ratios of the solvent and antisolvent, the particle size of the NPs can be controlled [99]. 

#### 3.1.2. Polymer Encapsulation

In general, polymeric NPs, such as poly(lactic-co-glycolic acid) (PLGA) and polylactic acid NPs, are synthesized using precipitation/solvent exchange/emulsification methods [100,101]. The double emulsion technique is commonly adopted to prepare NPs with polysaccharides and biodegradable polymeric materials [102].

For NP preparation, HPS are dissolved in an aqueous solution while the polymers are dissolved in dichloromethane; these solutions serve as the internal water phase and organic phase, respectively. The primary emulsion (water-in-oil) is obtained by completely mixing the two phases. Then, the primary emulsion is poured into the external water phase, and an emulsifier is added to generate a stable w/o/w emulsion [103]. After the removal of the organic solvent and centrifugation, NPs are obtained. Commonly used emulsifiers include F68 [104], F80 [105], Tween 20 [106], and triglycerides [107]. During the preparation process, the key factors that influence the quality of the NPs are the ratio of the organic phase to the internal water phase, the ratio of the external water phase to the organic phase, and the concentration of the emulsifier [108].

Polysaccharide NPs can also be synthesized using the microfluidic method. One approach involves the preparation of drug-loaded polysaccharide NPs that serve as the core and are coated with a shell made up of a polymer such as alginate [97]. In the other approach, a hydrophobic polymer, such as poly(D,L-lactide) (PLA), serves as the core, while an amphiphilic polysaccharide is employed as a stabilizer and surface modifier. Polysaccharide-covered NPs are prepared using a continuous emulsion/solvent diffusion method by employing a microfluidic flow-focusing junction [109].

#### 3.1.3. Covalent Link

Amphiphilic polysaccharide derivatives are widely known to self-assemble or aggregate in an aqueous phase. Briefly, polysaccharide-based micelles consist of three fundamental elements: the hydrophilic domain of the polysaccharides; the hydrophobic domain of the polymer; and the linkage between the hydrophilic and hydrophobic segments (Figure 1).

In general, there are two strategies for the synthesis of amphiphilic polysaccharide derivatives [110,111]. One involves the grafting of hydrophobic polymers with functional groups throughout the polysaccharide chains. In contrast, the other involves grafting the polymers onto polysaccharide terminal groups, generating “grafted” or “block” (or “block-like”) polysaccharides derivatives. For “grafted” structures, the synthetic polymers can be directly introduced through a “grafting onto” or “grafting from” approach. For “block” structures, polysaccharide derivatives are typically obtained via the “end-to-end coupling” of a preformed polymer and the polysaccharides. 

The inherent properties of amphiphilic polysaccharide derivatives, such as their Mw, degree of grafting, and volume ratio of hydrophilic/hydrophobic domains, greatly influence the physicochemical and biological properties of the derived NPs [112]. In particular, the degree of substitution (DS) of hydrophobic segments plays a key role in the quality of the micelles. As a general rule, a higher DS value is associated with a stronger hydrophobic force, which makes it easier for amphiphilic polysaccharides to self-aggregate into micelles. This results in a lower critical aggregation concentration(CAC) [113]. However, polysaccharides with too many hydrophobic modifications may precipitate in an aqueous solution, which reduces their drug loading capacity and encapsulation efficiency (EE) [114]. 

In addition, the linkage between HPS and hydrophobic segments is important for controlled drug release. This linkage is usually composed of stimuli-responsive bridges. Accordingly, these NPs disassemble in response to changes in internal and/or external stimuli, leading to the controlled release of loaded drugs [115,116].

*Grateloupia livida* polysaccharides-SeNPs are sometimes prepared using the selenite/Vc chemical reduction method by dissolving the polysaccharides in a redox system containing selenite and ascorbic acid [117]. The polysaccharides are capped onto NPs and produce a smoother NP surface. The particle size ranges from 50 to 200 nm, and the NPs show high stability and re-dispersion properties. 

#### 3.1.4. Crosslinking

In order to meet the special needs of medical application, HPS can be crosslinked to obtain three-dimensional hydrogels. In physically crosslinked hydrogels, different chains are linked through hydrogen bonding, ionic bonding, or an associative polymer–polymer interaction [118]. In contrast, in chemical crosslinking, covalent bonds/hydrogen bonds are generated between the polysaccharide and amino or carboxyl groups in other polymers by means of metal coordination [119] and the Schiff bases reaction [24]. Due to the formation of covalent bonds, chemical crosslinking provides greater resistance to environmental changes and stronger mechanical strength and stability than physical crosslinking. 

HPS-based metallic NPs (HPS-MNPs) are prepared through a crosslinking strategy. In general, the HPS are dissolved in an aqueous phase and added to metal salts, with continuous stirring at a specified temperature. The particles are precipitated after the addition of a large volume of ethanol and kept on an ice-bath overnight. After centrifugation and alcohol volatilization, the sediment is dissolved in water and then lyophilized to obtain HPS-MNPs [56,120]. The three key factors that influence the quality of the HPS-MNPs are the ratio of the polysaccharides solution to the metallic salt solution, temperature of the reaction, and pH of the reaction mixture [121,122,123].

Crosslinking strategies can also be used to prepare polysaccharide NPs without adding polymers [124]. In some cases, sodium trimetaphosphate (STMP) is employed as the crosslinking agent [125]. In these cases, the polysaccharides need to be modified to provide reaction sites for the crosslinking agent. *Araucaria heterophylla* polysaccharides [126] and *Prosopis chilensis* L. polysaccharides [127] can be formulated into NPs using this strategy. Before preparation, these polysaccharides can be carboxymethylated and then reacted with STMP to form NPs. These polysaccharides can be fabricated into curcumin-loaded nanocarriers with a size below 200 nm. The drug-loaded nanocarriers show both antibacterial and anti-cancer effects, along with a steady drug release profile [128].

#### 3.1.5. Physical Mixture

Physical modification is the most commonly used method to prepare HPS-DDS via non-covalent interactions between HPS and other polymers/drugs [129]. When mixed with other polymers, HPS can reduce viscosity and enhance mucosal adhesion. In addition, the carboxyl and hydroxyl groups provide hydrophilicity and degradability to the system [130]. For example, to prepare HPS microneedles, other polymers such as HA [49], CS [131], and alginate [132] are usually selected to improve the mechanical strength and skin puncture ability. In hydrogels, BPSs and carbomer are co-dissolved in an aqueous phase to form stable supramolecular aggregates and a cross-linked network structure [133]. Such a HPS–polymer mixture can significantly improve the colloidal strength and viscoelasticity of hydrogels [134]. 

#### 3.1.6. Electrostatic Adsorption

Some cationic polymers, such as CS [135] and polyethyleneimine (PEI) [135], can form nano-complexes with HPS through simple electrostatic interaction. In one study, to improve their intestinal absorption efficacy, APSs were mixed with amphiphilic CS to form nanocomplexes with a positive zeta potential and size of 100–150 nm. These nanocomplexes showed high permeation through intestinal enterocytes [136]. In another study, a CS-tamarind gum polysaccharides (TGP) [137] polyelectrolyte nanocomplex was fabricated to stabilize simvastatin and enhance its anti-tumor efficacy. The formulated NPs showed high simvastatin entrapment efficiency (67–83%) and a small size (53–383 nm). To improve water absorption and accelerate blood coagulation, alginate and BSPs were mixed with a CS solution to generate a CS/Alg/BSPs nano-complex via lyophilization [138].

### 3.2. Role of HPS in DDS

#### 3.2.1. Drug Carriers

HPS may serve as drug carriers in DDS, such as nanoparticles [139], HPS-drug complexes [140], hydrogels [141], and microneedles [142]. In such cases, HPS are introduced as a bio-copolymer or matrix to interact with the drug via absorption, conjugation, encapsulation, etc. [143]. Thus, biodegradability [10] and the compactivity between the HPS and drug [144] are the two key properties controlling drug release. The molecular weight [145], chain length [146], branches [147], charges [148], and special functional groups [149] of HPS affect the performance of the DDS. HPS with a low Mw can reduce the particle size of the NPs as well as the drug encapsulation efficiency, but increase the drug release rate [150]. In HPS-hydrogels, the chain length and branch properties significantly influence gel characteristics such as adhesion, mechanical strength, and rheological features, producing different drug release profiles [151]. 

#### 3.2.2. Macromolecular Drugs

As mentioned above, HPS produce immunomodulatory effect. When introduced into DDS, HPS can serve as both a drug carrier and a macromolecular drug for immunomodulation [32]. Hence, the simultaneous and synchronous delivery of bio-macromolecules and small drugs can be achieved through HPS-DDS. In this way, HPS-DDS provide more synergistic therapeutic effects through multiple levels and mechanisms [152]. Furthermore, in some HPS-based nano formulations, the in vivo stability and bioavailability of HPS can strongly be enhanced through molecular modification [153]. In previous studies, NPs were prepared using American ginseng polysaccharides for enhanced immunotherapy effects. *Lepidium meyenii Walp.* (maca) polysaccharides (MP) [154] and BSPs [26] were fabricated into micelles to prepare a co-delivery platform for the polysaccharides as well as small therapeutic agents. In this way, the specific targeting and cell absorption of the bio-macromolecules could be achieved. In another study, liposomes based on *Glycyrrhizophores* polysaccharides (GPSL) [155] and *Ophiopogon japonicus* polysaccharides [156] were found to show better immunomodulatory activities than the two HPS alone. In addition, the stability of *Ganoderma lucidum* polysaccharides could be improved significantly after their introduction into microcapsules [157].

#### 3.2.3. Adjuvants

With regard to immunotherapy, HPS are excellent immune adjuvants. They can: improve the levels of serum antibodies and promote the production of inflammatory factors such as IL-2, IL-4 and IL-6; promote the proliferation of lymphocytes; improve the abundance of CD4^+^ and CD8^+^ T cells in the peripheral blood; and thereby produce enhanced humoral and cellular immune responses [96,158]. Therefore, in HPS-based DDS, the polysaccharides often act as adjuvants and amplify the effects of vaccines [159]. For example, ASPs [160] and Chinese yam polysaccharides (CYP) [161] are used as adjuvants and co-encapsulated into PLGA NPs with ovalbumin (OVA). Owing to the presence of HPS, the immune responses induced by OVA are enhanced, as indicated by the higher ratio of CD4^+^ and CD8^+^ T cells as well as the long-term IgG immune response with a mixed Th1 and Th2 response. 

#### 3.2.4. Targeting Agents 

Many HPS are rich in galactose and mannose structural units and can specifically recognize the highly expressed asialoglycoprotein receptor (ASGPR) [162] and CD44 receptor [163] in tumor cells, as well as MRs [164], CD206 [165], and TLR-2 [166] in tumor-related macrophages. Hence, when incorporated into DDS, the HPS may serve as targeting agents for tumor cells and TME. For example, hepatoma cells can be specifically recognized by galactose, N-acetylgalactosamine, and glucose owing to the overexpression of surface ASGPRs [167]. Thus, given the large amount of galactose and glucose present in their structures, ASPs [168] and BSPs [169] can target liver tissue. Previously, ASPs and BSPs-based DDS have been developed for the targeted delivery of drugs to hepatoma cells. MP [170] have high levels of mannose, which shows specific affinity towards tumor-associated macrophages (TAM). Thus, they were used to deliver drugs to TME in our previous study. *Gracilaria lemaneiformis* polysaccharides (GLP) [171], which have a high binding affinity for the αvβ3 integrin overexpressed in glioma cells, have been used to prepare GLP–SeNPs DDS for glioma targeting. 

#### 3.2.5. Stabilizers

In recent years, numerous studies have reported that HPS can enhance the stability of nanosystems across a wide range of pH values and temperatures, and improve cellular uptake owing to their hydrophilic hydroxyl groups and high specific surface area [172]. In one study, the solubility of quercetin was increased by 68.88-fold after its covalent linkage to APSs, leading to enhanced stability. This was because the association constant between quercetin and the polysaccharides was high, resulting in a large solubilization coefficient in the system [173]. In SeNP systems, citrus lemon polysaccharides were used as modifiers and stabilizers [174]. With its high stability and good dispersion in water, the nanosystem showed significant antitumor effects against HepG2 cells. In HPS-Metal NPs (HPS-MNPs) [175] and HPS-inorganic NP systems [176], HPS possess excellent biocompatibility and provide effective transportation for MNPs in vivo. The presence of abundant -COOH and -OH groups in HPS can improve the solubility and stability of MNPs, resulting in a longer circulation time [177]. The polysaccharides reduce the toxicity of MNPs and increase their therapeutical effects and degradability. In addition, *Polygonatum sibiricum* polysaccharides [178] and APSs [179] have been introduced into SeNPs as stabilizers and dispersing agents to reduce their surface energy through hydrogen bonding, hydrophobic interaction, and electrostatic interaction.

#### 3.2.6. Emulgators

As a form of hydrophilic colloids, HPS have good emulsifying and thickening effects. Therefore, they are considered excellent emulsion stabilizers and are widely used in the preparation of emulsions [180]. In general, for the preparation of polysaccharide-emulsions, Tween-60, Tween-80, Span-60, and Span-80 are used as surfactants; soap, sodium lauryl sulfate, polyol fatty acid esters (such as monostearate glycerides), and polysorbates are mainly used as emulsifiers; and polyethylene glycol ethers, liquid paraffin, glycerin, soybean oil, olive oil, and stearyl alcohol are commonly used as the oil phase [181]. In particular, flavonoid-grafted soybean polysaccharides can emulsify unstable oil-in-water emulsions [182], and soy CS can be used to develop emulsion food delivery systems [183]. *Albizia lebbeck* L. polysaccharides (ALPS) are hydrophilic non-ionic polysaccharides. Owing to unique functional properties such as high thermal stability and pH sensitivity, ALPS show good potential as natural emulsifiers [181].

#### 3.2.7. Solubilizers

The solubilization effect of HPS is attributed to their specific structure and hydrophilic groups. The weakened intramolecular and intermolecular hydrogen bonds in HPS chains lead to low crystallinity and better water solubility [184]. Studies have shown that vinegar baked Radix Bupleurum polysaccharides (VBCP) can increase the solubility of baicalin and rhein. As natural surfactants, VBCP can self-assemble in water to form micellar aggregates, which can encapsulate insoluble drugs through the interaction of hydrogen bonds and hydrophobic forces [34].

**Table 2 pharmaceutics-14-01703-t002:** Summary of HPS-DDS.

Type of Dosage Form	Polysaccharides	Drug	Curing Disease	Preparation Method	The Role of HPS in DDS	Aimed Cells	Size (nm)	Zeta (mV)	Refs
HPS nanoparticles	*Panax quinquefolium* polysaccharides		Immunoregulation; Reduce skin cancer	Self-aggregation; Polymer encapsulation	Immunologic adjuvant; Macromolecular drugs	Macrophages	180 ± 10; 20		[99,185]
HPS–polymeric nanoparticles	Ramulus mori polysaccharides		Inflammatory bowel disease; Colitis	Polymer encapsulation	Macromolecular drugs	Macrophages	205.6 ± 4.26; 180.3 ± 4.21	−31.7 ± 1.097	[102,104]
	Inulin	Antigens	Stimulated the Th2 type immune response.	Polymer encapsulation	Immunologic adjuvant	Antigen presenting cells (APC)	1.5 ± 0.12		[105]
	Aloe polysaccharides		Angular leaf spot	Polymer encapsulation	Macromolecular drugs		644.00 ± 0.52; 243.20 ± 0.22		[106]
	Chinese yam polysaccharides	Ovalbumin	Strengthen immune responses	Polymer encapsulation; Covalent link	Immunologic adjuvant	CD3(+)CD4(+) T cells CD3(+)CD8(+) T cells	200		[186]
	*Angelica sinensis* polysaccharides	Ovalbumin; Inactivated H9N2	Induce strong and long-term immune responses; H9N2 influenza	Polymer encapsulation	Macromolecular drugs; Immunologic adjuvant	CD4(+)/CD8(+) T cells Th1 cells	225.2; 286.3 ± 2.45	−11.27; 47.8 ± 0.24	[187,188]
	*Dendrobium* polysaccharides	Ovalbumin	Improve immune responses	Polymer encapsulation	Immunologic adjuvant	Macrophages and lymphocytes CD4(+)/CD8(+) Tcells	141.4; 156.4; 175.9	−17.9 ± 1.29; −26.9 ± 2.76; 31.4 ± 2.18	[189]
Micelles	*Lepidium meyenii Walp(maca)* polysaccharides	Chloroquine	Cancer immunotherapy	Self-aggregation	Targeting; Immunologic adjuvant	4T1-M2 macrophages	120	−35	[154]
	*Angelica sinensis* polysaccharides	Curcumin; Doxorubicin	Acute alcoholic liver damage; Liver cancer	Polymer encapsulation; Covalent link	Drug carrier; Targeting	HepG2	208.4; 228	−20; −17	[162,168]
	*Bletilla striata* polysaccharides	Doxorubicin; Docetaxel; Silymarin; Let-7b; Alendronate	Antitumor; Liver diseases; Suppressive tumor microenvironment; Suppressed tumor progression	Crosslinking; Covalent link	Drug carrier; Targeting; Stabilizer	HepG2, HeLa, SW480, and MCF-7 HepG2 cell lines Michigan Cancer Foundation-7 (MCF-7) cells Dendritic cells (DCs) Macrophages	125.30 ± 1.89; 96.27 ± 1.21; 96.54 ± 5.27; 99.21 ± 3.83; 121.61 ± 9.81; 125.30 ± 1.89; 120; 67	−26.92 ± 0.18; −35.66 ± 0.28; −35.46 ± 0.10; −34.76 ± 0.22; −28.37 ± 0.12; −26.92 ± 0.18; −13; −19	[169,190,191,192,193,194,195]
	*Rehmannia glutinosa* polysaccharides	Inactivated Bb	*Bordetella bronchiseptica* (Bb)	Crosslinking; Covalent link; Self-aggregation	Immunologic adjuvant	Dendritic cells (DCs) CD4(+) and CD8(+) T-cells			[196]
	*Astragalus membranaceus* polysaccharides		Inhibited the growth of tumor	Self-aggregation	Immunologic adjuvant	Dendritic cells (DC) CD4(+) T/T-reg and CD8(+) T/T (reg)	138 ± 5	−12.4 ± 0.3	[197]
Polysaccharide-drug conjugations	*Lycium barbarum* polysaccharides	Platinum-based; Doxorubicin	Anticancer	Covalent link	Targeting; Immunologic adjuvant; Stabilizer; Immunologic adjuvant	A549 (human lung cancer cell line) human Hepatic cancer cell line HepG2	273.3	−25.6	[21,198]
	*Polygonum multiflorum* polysaccharides	5-fluorouracil	Antitumor	Covalent link	Targeting; Immunologic adjuvant	Splenocytes and peritoneal macrophages	124.7		[199]
Polysaccharide-Metal	*Rosa roxburghii* polysaccharides	AgNPs	Antibacterial	Crosslinking	Stabilizer		24.5-83.2	−36	[200]
	*Astragalus membranaceus* polysaccharides	AgNPs; AuNPs	Antibacterial; Antitumor and immunoregulation	Crosslinking	Stabilizer; Stabilizer; Immunologic adjuvant	Dendritic cells/T cells	65.08; 25.38	−28.33	[120,201]
	*Dioscorea opposita* Thunb polysaccharides	ZnNPs	Anti-diabetes	Crosslinking; Polymer encapsulation	Stabilizer; Immunologic adjuvant				[202]
	*Leucaena leucocephala* Seeds polysaccharides	AgNPs	Anticancer, Antifungal and Preservative	Crosslinking	Stabilizer; Solubilizer		8–20	−14.2	[203]
	Tamarind polysaccharides	AgNPs; AuNPs	Antibacterial; Anticancer and immunomodulatory	Crosslinking; Polymer encapsulation	Stabilizer; Immunologic adjuvant		44–86; 30–40; 20	−36.7	[204,205,206]
	Farfarae Flos polysaccharides	AgNPs	Anticancer	Self-aggregation	Stabilizer	HT29 cells	4–25	−17.1	[207]
	*Psidium guajava* L. leaf polysaccharides	AgNPs	Antioxidation or antimicrobial	Self-aggregation	Stabilizer		25–35	−25.23	[208]
	*Soybean* polysaccharides	AgNPs	Antibacterial	Self-aggregation	Stabilizer				[209]
	*Moringa oleifera* seed polysaccharides	AgNPs	Wound healing	Self-aggregation	Stabilizer		17.6	−25.6	[210]
	Apple polysaccharides	AuNPs	Anti-diabetes	Polymer encapsulation	Stabilizer		124 ± 8.55	−10.5 ± 0.54	[211]
HPS-inorganic nanoparticles	*Gracilaria lemaneiformis* polysaccharides	SeNPs	Anticancer	Covalent link; Crosslinking	Solubilizer; Targeting	Glioma cells	123	−24.0	[171]
	Citrus limon polysaccharide	SeNPs	Antitumor	Covalent link	Stabilizer		85.35; 79.67; 90.14	−9.44; −7.52; −6.87	[174]
	*Polygonatum sibiricum* polysaccharides	SeNPs	Antioxidation	Covalent link	Stabilizer		105	−34.9	[178]
	*Astragalus* polysaccharides	SeNPs; Chitosan	Antioxidation, enhance the proliferation of T-lymphocytes and Inhibit HepG2 cells proliferation; Exhibited high permeation through intestinal enterocytes	Covalent link; Self-aggregation; Electrostatic adsorption	Solubilizer	HepG2 cells	478.1; 100–150	−20.39; +16	[136,179]
	*Lignosus rhinocerotis* polysaccharides	SeNPs	Antioxidation	Covalent link	Immunologic adjuvant; stabilizer		50		[212]
	*Lycium barbarum* polysaccharides	SeNPs	Anti-fatigue; Antitumor; Protect human lens epithelial cells (HLECs) from UVB-induced damage; Antioxidation	Covalent link	Stabilizer; Solubilizer	Lens epithelial cells	105.4; 111.5–117; 150–200; 83–160	−37; −24.1	[111,213,214,215,216]
	*Codonopsis pilosula* polysaccharides	SeNPs	Inhibit the proliferation and promote apoptosis of HepG2 cells	Covalent link	Solubilizer; Stabilizer	HepG2 cells	75		[217]
	Dandelion polysaccharides	SeNPs	Anti-tumor	Covalent link; Polymer encapsulation	Immunologic adjuvant; stabilizer; Solubilizer	HepG2, A549, and HeLa	50		[218]
Other HPS based NPs	*Araucaria heterophylla* polysaccharides	Curcumin	Anticancer; Antioxidation and antibacterial	Physical mixture; Crosslinking	Solubilizer; Targeting; Drug carrier	MCF7 human breast cancer cell line	250–300; 200		[126,127]
	Tamarind Gum polysaccharides	Simvastatin	Antitumor	Covalent link; Electrostatic adsorption	Solubilizer; Targeting; Stabilizer	Human breast cancer cell line	53.3–383.1		[137]
Hydrogel	*Bletilla striata* polysaccharides	Lactobacillus plantarum probiotics	Skin lesions; Bleeding disorders; Wound infection	Covalent link; Crosslinking	Immunologic adjuvant; Stabilizer;	L929 cells			[30,219,220]
	*Plantago psyllium* seed polysaccharides	5-fluorouracil	Antitumor	Crosslinking	Immunologic adjuvant; stabilizer;				[221]
	*Tamarindus indica* polysaccharides	Silver nanoparticle	Wound infection	Crosslinking	Stabilizer				[222]
Microneedle	*Panax notoginseng* polysaccharides	Doxorubicin and 5-fluorouracil	Antitumor	Physical mixture	Drug carriers; Immunologic adjuvant	Skin dendritic cell			[49]
	*Bletilla striata* polysaccharides	Ovalbumin; Triamcinolone acetonide and verapamil	Infectious disease; Hypertrophic scars	Physical mixture	Stabilizer; Drug carriers; Immunologic adjuvant				[223,224]
HPS-based liposome	*Cordyceps sinensis Sacc* polysaccharides			Polymer encapsulation					[225]
	*Lycium barbarum* polysaccharides		Immunological and adjuvanticity against PCV2 in vivo	Polymer encapsulation	Immunologic adjuvant; Macromolecular drugs	Spleen cells, macrophagesCD4(+)/CD8(+) T cells	120.7 ± 0.84		[226]
HPS-Emulsion	*Dioscorea opposita* Thunb polysaccharides			Polymer encapsulation	Emulsifier		1500	−30	[107]
	*Albizia lebbeck* L. seed polysaccharides			Polymer encapsulation	Emulsifier		1160–2790	−35.83−−19.00	[181]
	*Soybean* polysaccharides	Soy protein	Emulsion digestion in the gastrointestinal tract	Polymer encapsulation	Stabilizer		835	−129.76	[183]

## 4. HPS-DDS and Their Immunomodulatory Effects

### 4.1. HPS-Nanomedicine 

#### 4.1.1. HPS NPs

AGPSs—which are well-known macromolecular immunomodulators—are also used to prepare polymeric NPs for enhanced immunotherapy [185,227,228]. AGPS NPs prepared using a microfluidic approach show an average size of 20 nm. These NPs are encapsulated within porous nanospheres (~180 nm) made up of biodegradable gelatin [229]. They show a concentration-dependent enhancement of immune stimulation in both cellular and animal models [227]. In addition, these AGPS NPs are also used for dermal application, as they can inhibit the ultraviolet (UV) radiation-induced imbalance in the endogenous antioxidation system owing to their anti-oxidative and anti-inflammatory properties [230,231].

#### 4.1.2. HPS–Polymeric Nanoparticles

In previous studies, HPS with strong immunoregulatory activities, such as *Ramulus mori* polysaccharides (RMPs) [102,104], *Amomum longiligulare* polysaccharides (ALPs) [100], and CYPs [101,186], have been encapsulated into PLGA to form HPS-PLGA NPs using the double emulsion method. The obtained NPs have a spherical morphology and a narrow size distribution. In addition, the particle size varies from 100 nm to 400 nm depending on the type of polysaccharides and the preparation process used. 

In polymeric NPs, the polysaccharides can serve as immunoregulatory biomacromolecules. Some HPS can downregulate the expression of inflammatory cytokines while promoting the production of IL-10 and boost the phenotypic switch from M1 to M2 in macrophages. For example, RMPs were packaged in PLGA NPs to develop a novel anti-inflammatory nanomedicine and treat acute inflammatory bowel disease by regulating macrophages and T cells [102,104]. In contrast, some HPS, such as ALPs and CYPs [186], can re-polarize M2 macrophages to the M1 phenotype and produce a strong immune-activation effect. Further, both these contrasting immune regulation activities can be enhanced through nanosizing and polymer encapsulation. 

Despite their therapeutic potential, the low bioavailability and brief biological half-life of HPS have limited their clinical applications. HPS–polymeric NPs represent an effective tool for improving the in vivo stability and bioavailability of HPS. Accordingly, the immunoregulatory activities of HPS are significantly enhanced after encapsulation into NPs [100,101]. 

In addition, these NPs serve as a platform for the co-delivery of polysaccharides and vaccines. In these applications, the polysaccharides act as adjuvants and amplify the effect of vaccines. For example, in one study, ASPs and OVA were co-encapsulated into PLGA NPs [187]. Because of the presence of ASPs, the immune responses induced by OVA were enhanced, as indicated by the higher ratio of CD4^+^ to CD8^+^ T cells as well as the long-term IgG immune response with a mixed Th1 and Th2 response. In another study, PEI-modified CYP-encapsulated PLGA NPs (CYP-PEI) were developed as a potential vaccine delivery system to trigger strong and persistent immune responses [232]. 

Stimulus-responsive HPS–polymeric NPs can also be developed. ASPs-ammonium bicarbonate co-encapsulating PLGA NPs [103] were prepared in one study. Here, the ammonium bicarbonate conferred a pH-responsive effect to the delivery system. Accordingly, ASPs release was boosted. 

HPS-PLGA NPs can be further developed to achieve a complex nanostructure. In earlier works, CYP-PLGA NPs and ASPs-PLGA NPs [188] were surface modified using PEI. These positively charged PEI-coated NPs enabled the delivery of the antigen and CYP immunomodulator to lymph nodes and activated DCs, further enhancing the mixed Th1/Th2 response and Th1-biased immune response in vivo. In another study [189], dendrobium polysaccharides (DP)-PLGA-OVA NPs were developed and coated with a PEI-modified macrophage cell membrane. The macrophage cell membrane enhanced the immune effectiveness, while the PEI provided a positive charge and improved cell adsorption. Accordingly, biomimetic nano-vaccines can promote antigen uptake by macrophages and enhance lymphocyte proliferation.

### 4.2. HPS Amphiphilic Derivatives 

#### 4.2.1. HPS-Based Micelles 

The development and applications of polysaccharide micelles as drug carriers are widely reported, BSPs are a good example. Certain hydrophobic polymers can be grafted onto the BSPs to form amphiphilic polysaccharides-copolymer derivatives using compounds such as cholesteryl [233], stearic acid (SA) [113,169,190,191], fatty acids [110].

The prepared amphiphilic polysaccharide derivatives can be self-assembled into micelle structures with particle sizes of 100–300 nm depending on the proportion of the hydrophilic and hydrophobic domains. Given their good performance in hemolysis and cytotoxicity analyses, these micelles can be used as drug carriers. Hydrophobic drugs, including docetaxel [191] and doxorubicin (Dox) [26,113,192], can be encapsulated into the hydrophobic core of these micelles. In addition, other HPS, such as APSs [162,168], and *Rehmannia glutinosa* polysaccharides (RGPSs) [196] can also be used to prepare micelles with the “grafted onto” strategy. 

Through further structural modification, stimuli-responsive properties can be added to the micelles. Typically, two strategies are used for such modifications. In the first strategy, functional groups are directly introduced onto the polysaccharide chain. In our previous study, histidine (His)- and SA-modified BSPs derivatives were synthesized. The prepared micelles showed stepwise pH sensitivity due to the protonation-deprotonation effects of the imidazole group in His. Therefore, the prepared BSPs micelles showed boosted drug release under acidic conditions [26,192]. In the second strategy, the functional group is used as a linker. For example, one study used cystamine (CYS) to link BSPs and SA. As the CYS linker could be broken down under reducing conditions, the micelles could be disassembled, resulting in a fast drug release profile [234]

Micelles can also be modified to achieve targeting effects. In a previous study, a folate-and SA-modified BSPs (FA-BSPs-SA) polymer was synthesized. The micelles could enter tumor cells through folate receptor-mediated endocytosis via a clathrin-dependent mechanism. This facilitated the synergistic anti-tumor effects of the loaded Dox in 4T1 cells, resulting in enhanced anti-tumor effects [193]. 

Owing to their metabolic features, HPS micelles possess some unique properties. It has been reported that hepatocellular carcinoma cells can specifically recognize various ligands such as galactose, N-acetylgalactosamine, and glucose via the ASGPR expressed on their surfaces [235] Thus, ASGPR-mediated DDS have gained interest in the field of hepatocellular carcinoma treatment. It was demonstrated that some HPS, such as BPSs and ASPs, show a high affinity to ASGPR due to their higher galactose content, branched structure, and appropriate spatial geometry. Hence, BSPs and ASPs micelles possess liver-targeting capabilities and are promising nano carriers for drug delivery to the liver [168,169]. In addition, the specific glycosidic linkages of polysaccharides can also confer targeting properties. BSPs contain β-glucose and α-mannose at a relative molar ratio of ~2.4:1. MRs are strong lectins that can recognize and bind to BSPs. Due to the overlapping expression of MRs on TAMs and tumor-infiltrating dendritic cells (TIDCs), BSPs nano vehicles can be used as targeted DDS for tumor immunotherapy [194,195]. 

HPS micelles can also be used as a co-delivery platform for bioactive macromolecules and small therapeutic agents. In a previous study, we fabricated TAMs-targeting amphiphilic polysaccharide micelles for synergistic cancer immunotherapy. We grafted a PLGA segment onto MP, which are naturally derived macromolecules with a strong TAMs-remodeling effect. Disulfide bonds were used as a redox-sensitive linkage. The amphiphilic polysaccharide derivatives could self-assemble into core (PLGA)-shell (MP)-structured micelles and encapsulate chloroquine (CQ) into the hydrophobic core. After administration, the micelles co-delivered MP and CQ to TAMs and achieved a synergistic immunotherapeutic effect in tumors via the multiple regulatory effects of MP and CQ. Accordingly, the targeted delivery of active macromolecules was achieved [154]. In another study [197], the hydrophobic compound 4-(n-octyloxy) phenylboronic acid was simply conjugated to APSs. The amphiphilic polysaccharide derivatives could self-assemble into NPs (ANPs). These ANPs could reverse TEM and thereby enhance the radiation-induced abscopal effect. A mechanistic study demonstrated that the ANP-induced immune response was mainly mediated by DCs activation, manifesting as phenotypic maturation and enhanced antigen presentation through the TLR4 signaling pathway.

#### 4.2.2. HPS-Drug Conjugations

Polymer-drug conjugates (PDC) have exhibited clinical and commercial success in both drug delivery and targeted treatment for cancer, diabetes, arthritis, and pain [236,237,238]. At present, polysaccharides-drug conjugates (PSDCs) appear to be effective options and have attracted considerable attention. PSDCs are simply prepared by covalently linking a drug to the functional groups in the polysaccharides. During the preparation process, the reaction time, reaction temperature, amount of catalyst, and molar ratio of polysaccharide to drug affect the drug loading activity [239].

HPS can also act as bioactive agents. Some immunomodulators—such as ASPs, *Lyc**ium barbarum* polysaccharides (LSPs), *Polygonum multiflorum* polysaccharides (PMPSs), and APSs—were employed to construct PSDCs [34]. The PSDCs enabled the co-delivery of these bioactive macromolecules with therapeutic agents such as dexamethasone (Dex), platinum-based antineoplastic drugs, Dox, 5-fluorouracil, and ibuprofen, among others [240,241]. Due to their self-assembly characteristics, the prepared PSDCs possessed the specific features of NPs, showing selective accumulation in reticuloendothelial system phagocytic cells as well as tumor cells. The LBPs-Dox complex showed the strongest inhibition against the proliferation of tumor cells, higher than that of polysaccharides and the drug alone, indicating that LBPs and Dox exerted a synergistic anti-tumor effect [21]. When 5-fluorouracil was covalently linked to PMPSs, the levels of IL-2 and TNF-α were further upregulated. Thereby, the immunosuppression in TME was re-sharped, and the anti-tumor effects were enhanced [199]. The ASPs-Dex conjugate has also been shown to exert a synergistic effect in the treatment of ulcerative colitis in rats. The ASPs limit the overall amount of drug absorbed and thus reduce the systemic immune suppression caused by Dex alone [242].

In addition, the high fractions of mannose and glucose in PSDCs enable targeted delivery to the liver by resident macrophages and sinusoidal endothelial cells. Due to the overlapping expression of MRs on TAMs and TIDCs, cationic BSPs (cBSPs), which contains high amounts of mannose moieties, can be used to conjugate therapeutic agents such as bisphosphonate and let-7b and efficiently target and specifically re-shape TME for cancer immunotherapy [194,195]

### 4.3. Polysaccharides-Metal Nanoparticles

Metallic NPs (MNPs) are promising materials for DDS [243]. HPS have been recognized as valuable agents for the synthesis of MNPs owing to their outstanding biocompatibility, biodegradability, and targeting properties [244].

Generally, HPS-MNPs exhibit a “core-shell” structure [202]. The core is composed of metallic NPs (Pt, Au, Ag, Zn, or their oxides), while the polysaccharides are coated onto their surface, acting as both stabilizing and reducing agents and adhering to the MNPs through noncovalent bonding [203,204,207]. The carboxyl and hydroxyl groups of the polysaccharides play a key role in mediating the solubility and stability of the MNPs in the HPS-MNP complex, preventing the flocculation and aggregation of MNPs [245]. As a result, the HPS-MNPs complex can form a stable suspension that carries a negative electrostatic surface charge (Figure 2). Moreover, the particle size ranges from 10 to 150 nm [201,246]. 

In some cases, HPS such as APSs [120] and carboxymethyl tamarind polysaccharides (CMTs) show synergistic antibacterial effects along with MNPs. It has been reported that CMT AgNPs (CMT-AgNPs) exert anti-bacterial effect without damaging immune cells such as macrophage and keratinocytes. Hence, they hold outstanding promise in the treatment of bacterial infections in vivo [205]. Other polysaccharides, such as *Rosa roxburghii* Tratt fruit polysaccharides (RP3) [200], and soluble soybean polysaccharides (SPSSs) [209], do not have significant antibacterial effects when used alone, but can strongly enhance the activities of MNPs, enabling sustained release with long-term use. 

During the healing process, the polysaccharides can regulate macrophages, which play an important role in cell proliferation, vascularization, collagen deposition, and granulation, by secreting various chemokines and cytokines. *Moringa oleifera* seed polysaccharides-embedded silver NPs (MOS-PS-AgNPs) show a greater bactericidal effect and lower toxicity than plain AgNPs. In addition, the complex promotes wound healing by upregulating IL-10 and downregulating IL-6 expression [210]. 

Further, PMNPs can target different tumor cells and treat various types of cancers depending on the polysaccharides and MNPs used. The polysaccharides PST001 are isolated from the seed kernels of *Tamarindus indica* (PST), and PST-Gold NPs exert anticancer effects through the induction of apoptosis and increase the count of CD3/4/8 cells in the bone marrow [206]. One study showed that ASPs-AuNPs can increase not only NO release in the culture environment of DCs but also enhance the gene expression of DCs-derived cytokines, promoting the proliferation of CD4^+^ and CD8^+^ T cells in splenocytes [201]. The Farfarae Flos polysaccharides-Ag complex (FFP@AgNPs) significantly decreases the proliferation ability of HT29 tumor cells, inhibits their migration, and promotes their apoptosis [207]. In LBPs platinum-based conjugates (LBPs-5-ASA-Pt), Pt shows better binding to DNA and forms Pt-DNA adducts, leading to greater nuclear accumulation and irreversible lesions in the double helix, which ultimately result in tumor cell death [247]. 

Further, HPS-MNPs have been developed as new potential candidates for the treatment of diabetes mellitus (DM) and its complications. Modified apple polysaccharides (MAP)-AuNPs were conjugated with insulin (INS) for the oral treatment of type 1 DM [211] while a *Dioscorea opposita* Thunb. polysaccharides-zinc (DPS-Zn) inclusion complex was synthesized for the treatment of type 2 DM [202]. In this formulation, the HPS can restore the immune balance by regulating cytokines and enhancing the body’s immunity, producing synergistic treatment effects against DM.

### 4.4. HPS-Inorganic Nanoparticles

Selenium (Se) is an essential micronutrient that exerts its biological effects mainly via selenoproteins. SeNPs have functional characteristics such as a high absorption rate, low toxicity, and high biological activity. Therefore, they are widely used in health foods and the field of medicine. However, SeNPs have the disadvantages of a high surface energy and poor stability. They easily form grey and black elemental selenium and lack selectivity against cancer cells. Considering that both HPS and SeNPs possess unique physicochemical characteristics and bioactivities, it is reasonable to assume that HPS-SeNPs would have their combined advantages and show stronger biological effects, such as anti-tumor, anti-oxidation, and immunomodulatory effects. Thus, they could potentially be used as nutritional Se supplements in food and medical applications. 

When LBPs are used as the capping agent, LBPs-SeNPs exhibit better stability [141], with enhanced bioactivities, including anti-fatigue [214], anti-tumor [111], UV Protection [215], and antioxidation effects [216]. *Rosa roxburghii* fruit polysaccharides (RTFP)-SeNPs exert synergistic protective effects against H_2_O_2_-induced apoptosis in INS-1 cells. In this formulation, RTFP not only serves as the stabilizer but also exhibits high antioxidation and α-glucosidase-inhibiting effects. Inulin, a fructan-type HPS widely distributed in nature, has been demonstrated to have prebiotic effects and anti-tumor, anti-oxidation, and immunomodulatory activity. In a previous report, inulin fructans from *Codonopsis pilosula* [248], *T. mongolicum* [218] as well as *Citrus limon (L.) Burm. f.* (Rutaceae) [174] were introduced into SeNPs for tumor treatment. In addition, *Ulva lactuca* polysaccharides-SeNPs were found to potentially attenuate colitis by inhibiting NF-κB-mediated hyperinflammation. 

### 4.5. HPS-Based Hydrogels

Hydrogels are physically or chemically cross-linked hydrophilic polymers that can immobilize large amounts of water or biologic fluids into their 3D structures [249]. The polymeric network in hydrogels consists of hydrophilic polymer chains that are linked to each other via physical and chemical crosslinking [250]. HPS can be introduced into hydrogel systems to improve their biocompatibility. Polysaccharides in hydrogels can play three major roles. First, they can act as a pharmaceutical agent and exert independent therapeutic effects. Second, they can enable more functional modifications and respond to a greater variety of biological environments. Finally, the reducing groups of polysaccharides can improve the stability of metal NP-hydrogel composite systems and play a protective role.

The simplest approach for hydrogel preparation involves physical cross-linking (one-pot method). Currently, BSPs and APSs are commonly used to prepare hydrogels with good biocompatibility and mechanical strength via physical crosslinking. A *Snakegourd* root/*Astragalus* composite OPCH hydrogel was successfully printed into three differently shaped patches using a melt extrusion 3D printer [89]. Previously, we prepared a BSPs hydrogel using the one-pot method with carbomer as the matrix. In another work, BSPs was mixed with the hydrophilic monomer acrylic acid (AA) [30], and the linear polymer polyvinyl alcohol (PVA) was introduced to create a double network [219]. To enhance the mechanical properties of the hydrogel system, CS was reacted with oxidized BSPs (OBSPs) [220], resulting in greater intermolecular bonding and better wound healing activity [251] from an immune regulation perspective. 

Furthermore, chemical- and radical-induced free radical polymerization can also be used to prepare HPS-hydrogels. For example, the graft copolymer hydrogels and three-dimensional interpenetrating networks (IPNs) of *Psyllium* polysaccharides (Psy) and methacrylic acid (MAAc) were prepared using ammonium persulphate as the initiator and N,N-methylenebisacrylamide (N,N-MBAAm) as the crosslinker [252,253]. In addition, the irradiation of polymers in a suitable system is also useful for preparing hydrogels without any chemical initiator/cross linker [254]. *Psyllium*-N-vinylpyrrolidone (NVP)-based hydrogels (psy-cl-poly [NVP]) were prepared through radiation-induced crosslinking [221]. The polymerization involved the irradiation of an aqueous *Psyllium* polysaccharides solution, resulting in the generation of a significant number of radicals on polymer chains [221]. 

In particular, composite polysaccharide hydrogels loaded with metal NPs have also received attention. In these systems, the metal NPs act as antibacterial agents, while the HPS serve as a carrier for the metal NPs and reduce toxicity and promote wound healing. Galactoxyloglucan (PST), a HPS isolated from the seed kernel of *Tamarindus indica*, was used to prepare hydrogels with Au/Ag NPs [222].

### 4.6. HPS-Based DMNs

Dissolving microneedles (DMNs) have been widely studied for their applications in transdermal drug delivery because they can dissolve within the skin [223]. HPS are extensively used for the fabrication of DMNs due to their biocompatibility, biodegradable nature, and sustainable delivery. The most widely explored polysaccharides for DMN-related applications include HA, dextran, CS, cellulose, sodium alginate (SA), and blends of other biopolymers [255]. Compared with ordinary polymers, HPS not only offer a wider range of functional groups, good biocompatibility, biodegradability, and other polymer-related properties, but also have immunomodulatory action (Figure 3).

BSPs microneedles (BMNs) are the most widely studied. Mechanically robust simple BMNs are prepared using centrifugation [142,223]. The prepared BMNs exhibit better mechanical properties and stability than microneedles made of HA and polyvinyl alcohol. BMNs can dissolve in the interstitial fluid of the skin after insertion and release the encapsulated OVA to prime the immune system. These encouraging findings indicate that BMNs can be a promising tool for effective vaccine delivery. Further, the micro-molding method can be used to prepare more complex forms of MNs with carboxymethyl CS (CMCH) and BSPs for hypertrophic scar (HS) treatment [224]. Hydroxypropyl β-cyclodextrin (HP-β-CD) is used to encapsulate hydrophobic triamcinolone acetonide (TA), and the obtained inclusion is co-loaded with hydrophilic verapamil (VRP) in MNs. The MN-HP-β-CD(TA)-VRP MNs are then attached to an EC-based layer to obtain a MN patch, which shows a higher mechanical strength and stronger internal interaction of hydrogen bonds than observed with BSPs alone. The BMNs significantly decrease the thickness of HSs as well as their hydroxyproline (HYP) and transforming growth factor-β1 (TGF-β1) expression. Accordingly, they improve collagen fiber arrangement and reduce dermis congestion and hyperplasia.

In our previous study [49], we prepared HPS MNs using Panax notoginseng polysaccharides (PNPS) as the matrix. The PNPS MNs possess suitable mechanical strength and good solubility, enabling them to easily penetrate the SC layer and release the loaded drug into the deeper layers of skin. Upon entering the skin, the bioactive PNPS recognize and target skin DCs via TLR2 and TLR4, triggering DCs maturation and enhancing the local DCs-mediated T cell immune response. Thus, these PNPS MNs could serve as promising carriers for transdermal drug delivery and natural adjuvants for transcutaneous immunization.

### 4.7. Other HPS-Based DDS

#### 4.7.1. Liposomes

The development of HPS-lipid formulations has become a hot spot of research. Liposomes can encapsulate polysaccharides in their aqueous cores and/or membranes, and the unique physical and chemical properties of liposomes not only improve the bioavailability, therapeutic effect, and dosage of polysaccharides but also enable their targeted delivery [256,257,258]. 

The polysaccharides commonly used in the preparation of liposomes are white mustard polysaccharides, Yu ping feng polysaccharides, wheat dong polysaccharides, APSs, and goji berry polysaccharides. In addition, valuable herb plants such as *Ganoderma lucidum* and *Cordyceps sinensis* are also used in the development of polysaccharides-liposomes. Compared with APSs, APSs-liposomes not only induce greater phagocytic activity in macrophages but also improve the ability of DCs to stimulate T cell proliferation and develop antigens [225]. Wheat dong polysaccharides-liposomes can significantly increase SOD and oxidase levels and reduce MPO levels. Further, they can significantly increase the proportion of splenocyte proliferation and CD4^+^ and CD8^+^ T cells while also enhancing cytokine secretion [259]. LBPs liposomes can act in synergy with PHA or LPS to significantly promote splenocyte proliferation, increase the ratio of CD4^+^ and CD8^+^ T cells, promote macrophage cytokine secretion; enhance the PCV2-specific IgG antibody response, and promote the secretion of IFN-γ, TNF-α, and IL-4 [226].

#### 4.7.2. Emulsions

Some HPS, such as *Albizia lebbeck* L. seed polysaccharides, *Dioscorea opposita* Thunb polysaccharides, and APSs can be used as natural emulsifiers and smart polymers for the preparation of pH-sensitive drug delivery system emulsions [107,181]. In addition, APSs nano-emulsions can induce specific antibodies and enhance the Th1 and Th2 immune response. They can significantly enhance the body’s ability to produce antibodies and enhance the immunogenicity of antigens, improving vaccine effectiveness and reducing the immune response after vaccine administration [107].

## 5. Conclusions and Prospect

HPS are biodegradable, carbon neutral, environmentally friendly, and pose a low risk to human health and safety. They are easily modifiable and have other unique physical and chemical properties. Therefore, they are good carriers for DDS. HPS also have immunomodulatory effects, which further add to their advantages. Therefore, the application of HPS in DDS has become a research hotspot. In this review, a wide range of HPS-DDS were systematically summarized through formulation-based categorization. A brief introduction of some frequently studied HPS was provided, followed by strategies and methodologies used for polysaccharides-based DDS fabrication. In particular, we focused on the role of HPS in these DDS.

Many studies have been reported that HPS are rich in biological activities, safe and non-toxic side effects. However, HPS-DDS has not been deeply discussed. Although some clinical studies on HPS-DDS have been initiated, research on HPS-DDS is largely limited to the preclinical stage, and some problems need to be addressed before their clinical translation, such as the evaluation of their long-term safety, toxicological profile and the quality control standard system and so on. Further, Novel characterization and preparation techniques are required for controlling the quality of HPS-DDS and overcoming batch variations in the polysaccharides. In addition, scalable, cost-effective, and reproducible manufacturing systems need to be developed for HPS-DDS. In addition, the interactions between the polysaccharides and other components of the DDS as well as the mechanisms of polysaccharide NPs synthesis need to be investigated more thoroughly. In view of the above problems, HPS-DDS still has many potential problems before they get approval by regulators. Therefore, it is still needing the unremitting efforts of scientific researchers to establish the more perfect structure of HPS-DDS.

As a natural product, HSP were obtained by extraction and purification from herb plants. Thus, the purity was an important issue influencing the functions. For example, trace amounts of lipopolysaccharides in polysaccharides would significantly affect their immunoregulatory activities. In the relevant articles, the HSP used in DDS are obtained under complex purification process. First, the HSP should be deproteinization and decolorization in advance. Then, the HSP were purified by DEAE column to remove the general impurities (e.g., oligosaccharides, proteins and lignin). After that, they are further purified by sephadex column including G20, G50, G100 and G200 to obtain the homogeneous products with narrow Mw distribution. As a result, the products usually have an acceptable purity above 90%. In our previous work, PNPS, MP and BSPs were used to fabricate the DDS such as micelle, microneedles and hydrogel. the purities for all there HSP were all above 95% [49,154,192,260].

When designing HPS-based NPs for a specific purpose, it is crucial to control polysaccharides properties such as molecular weight, monosaccharide composition, and type of glycosidic linkage. However, unlike synthetic polymer materials, HPS are not amenable to property modifications. The region of plant production, plant age, and pretreatment methods all significantly influence the quality of HPS. In addition, preparation methods such as water extraction, alcohol precipitation, and column chromatography, also affect the properties of HPS. Therefore, the internal relationship between the preparation process and the quality of polysaccharides needs to be further explored. Moreover, integrated extraction and separation technologies need to be employed at a large scale to standardize the extraction and purification process. In this way, a large number of raw polysaccharide materials could be obtained for further investigation while controlling quality.

Numerous studies show that many HPS have significant immunomodulatory activity and can inhibit tumors by independently activating the immune response. This is a major advantage of HPS-DDS. However, the bioactivity of polysaccharides is closely related to their structure. Thus, caution must be exerted before structurally modifying HPS during the fabrication of DDS. Physical modification is more reliable and is less likely to alter the original structure of the polysaccharides. However, chemical modifications, which are sometimes performed to obtain polysaccharide derivatives, can disrupt HPS activity. In such cases, the structure–activity relationship of polysaccharides should be taken into consideration, as the active sites of the polysaccharides could change after modification. Specifically, in some cases, the polysaccharide chain can break down after modification, leading to a loss or change in function. Therefore, reasonable and efficient chemical modification strategies are necessary.

Despite the significant efforts made towards understanding the potential of HPS in DDS, several aspects remain to be elucidated. However, given the primary focus on evaluating the influence of different hydrophobic modifications, the effectiveness of different combination modes and sites of polysaccharides and polymers/drugs has received limited attention. Moreover, the structure of HPS–polymer derivatives needs to be evaluated further to understand the related self-assembly mechanism. In summary, to fabricate HPS-based NPs with desirable and controllable properties for biomedical applications, the interactions between HPS and other components should be comprehensively analyzed in future studies.

## Figures and Tables

**Figure 1 pharmaceutics-14-01703-f001:**
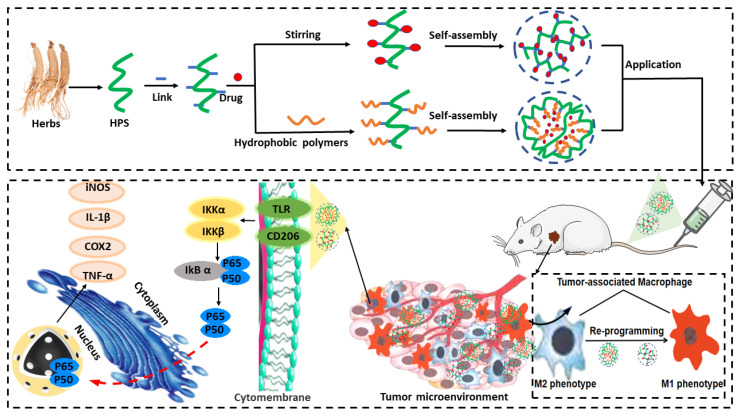
Fabrication and application of HPS amphiphilic derivatives.

**Figure 2 pharmaceutics-14-01703-f002:**
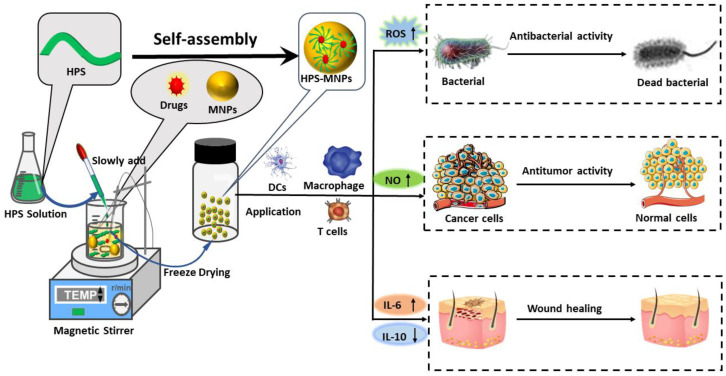
Fabrication and application of HPS-MNPs. (↑: Increase; ↓: Decrease).

**Figure 3 pharmaceutics-14-01703-f003:**
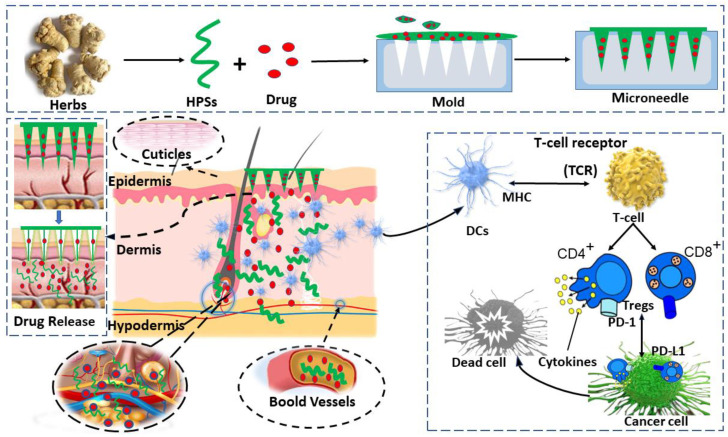
Fabrication and applications of HPS-based DMNs.

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
