# Peer review of "Herb Polysaccharide-Based Drug Delivery System: Fabrication, Properties, and Applications for Immunotherapy"

_pharmaceutics, 2022, doi:10.3390/pharmaceutics14081703_

Round 1

Reviewer 1 Report

In the presented review, the authors collected a large amount of material, but the idea of ​​the review is not clear. The introduction does not give enough information of ​​the reasons for choosing the purpose of the review, it is not structured, there are no data on similar reviews with clear evidence that is already known on this topic. It is totally unclear why  immunomodulatory activity is included here. Article includes enumeration of the various preparation of DDS but no logic observed in total. Also not all polysaccharides known mentioned in the text for example, no data on arabinogalactan. It is not clear why the authors separate HPS from PPS? There are many statements in the text with no references to literature. In general, the review looks raw and not systemic. There are many mistakes and missprints. Here are some examples:

Line 31 – link after chem. properties

Line 61 – no link

Line 63 – HA and dextran in italic

Line 64 – PS, what is this?

Line 76 – it’s a repeat from introduction

Line 79 – “of these …. action” No proof.

Info in Lines 89-94 is similar to 100-104

Line 103-108, 117-120 - why this info is in the Immunomodulation section?

Table 1 cited in the text as for Immunomodulation but in fact consists a lot of additional info.

Table 2 column Polysaccharides include names of the origin but not Polysaccharides’ names.    

Section 3.1. – why nanomedicine?

So, in the present form this review is not acceptable for publication. 

Author Response

Ms. Ref. No.:pharmaceutics-1803570

Title: Herb Polysaccharide-based Drug Delivery System: Fabrication, Properties, and Applications for Immunotherapy

Dear reviewer 1:

Thank you for your letter and the kind comments concerning our manuscript. We have studied comments carefully and have made appropriate and acceptable revisions. The revised parts were marked in red in the manuscript. The main corrections in the paper and the responds to the comments are as following:

  1. In the presented review, the authors collected a large amount of material, but the idea of ​​the review is not clear. The introduction does not give enough information of ​​the reasons for choosing the purpose of the review. it is not structured, there are no data on similar reviews with clear evidence that is already known on this topic. It is totally unclear why immunomodulatory activity is included here. Article includes enumeration of the various preparation of DDS but no logic observed in total. Also not all polysaccharides known mentioned in the text for example, no data on arabinogalactan. It is not clear why the authors separate HPS from PPS? There are many statements in the text with no references to literature. In general, the review looks raw and not systemic.

Response: The authors appreciated for the kind suggestions of the reviewer. According to the comments, we have clarified the reasons for choosing the purpose of this review in the introduction (section 1: pages 2-3). It was widely accepted that some Natural polysaccharides, such as chitosan (CS), alginate and hyaluronic acid (HA) and etc, are good biomaterials for fabrication of DDS. Among them, HPS is a kind of nature PS derived from medicinal plant resources and featured by their strong immunomodulatory activity. Therefore, In addition to good biodegradability, HPS-based DDS also have advantages of immunotherapy. Therefore, this review focuses on the recent research progress of HPS-DDS in combination with its immunomodulatory activity and biomaterial properties.

According to the comments, the authors made a thorough revision to make the paper logical and specific. Firstly, the immunomodulatory activity of mentioned HPS was summarized in section 2, and the potential mechanism was discussed from the perspective of structure-activity relationship. Second, in section 3, the principle and strategy of fabrication of HPS-DDS was described. Then, the role of HPS in DDS was summarized with emphasizing the immunomodulatory effects. Based on this, we reviewed the recent research progress of classical HPS-DDS formulations such as nanoparticles, micelles, hydrogels, microneedles in section 4. Finally, we point out the main limitations of HPS-DDS and proposed the further study directions.

  1. There are many mistakes and missprints. Here are some examples:

Line 31 – link after chem. Properties

Response: Thanks for your kind suggestions, which is valuable for improving the accuracy of the manuscript. We have modified this issue and added corresponding references, as seen in page2, lines 62-66.

Line 61 – no link

Response: Thank you for your references, which are now included in the revised manuscript. We have explained the problem you raised in this paper, and used previous studies to illustrate this problem, and also added corresponding references, as seen in page 3, lines 98-106.

Line 63 – HA and dextran in italic

Response: Thanks for your kind suggestions, we have modified this issue and corrected formatting issues., as seen in page 2, lines 66-67.

Line 64 – PS, what is this?

Response: This observation is correct, we have changed, PS is the short form of polysaccharide, we have all modified to HPS.

Line 76 – it’s a repeat from introduction

Response: This manuscript has been revised extensively according to the reviewers' constructive suggestions.as seen in page 2, lines 55-57

Line 79 – “of these …. action” No proof.

Response: Thanks for the references, which are now included in the revised manuscript. Specific references are listed as follows as seen in page 3, lines 79-81.

Info in Lines 89-94 is similar to 100-104

Response: Thank you for your constructive suggestions, we reviewed the manuscript carefully and revised it accordingly, including some duplications.as seen in page 4, lines156-161

Line 103-108, 117-120 - why this info is in the Immunomodulation section?

Response: We have modified the article to reflect the relationship between polysaccharides and their immunomodulatory activity.as seen in page 3, lines121-127

Table 1 cited in the text as for Immunomodulation but in fact consists a lot of additional info.

Response: We have modified Table 1 to appropriately remove additional information content and to fully reflect the relationship between polysaccharide structure and biological activity.

Table 2 column Polysaccharides include names of the origin but not Polysaccharides’ names. 

 Response: We have modified the source of polysaccharides in Table 2 to the name of polysaccharides, and added various preparation methods of HPS-DDS, and slightly modified "The role of HPS in DDS". 

  1. Section 3.1. – why nanomedicine?

Response: we have re-structured this section (section 4). “nanomedicine” were divided from other HPS based nano formulations. Further, we classified “HPS nanoparticles” and “HPS-polymeric nanoparticles” into “nanomedicine”. In the two formulations, HPS were mainly served as macromolecular drugs and were nano-sized for enhanced their delivery and absorption profiles in vivo, while in other nano DDS formulations, the HPS also exhibited additional functions such as drug carriers and stabilizer.

Overall, we thank the kind comments. Those comments are all valuable and very helpful for revising and improving our paper, as well as the important guiding significance to our researches. Furthermore, a language editing services was employed. We hope now the manuscript can meet the publication standards of the journal after our corrections.

Name: Chengxiao Wang

Unit: Kunming University of Science and Technology

E-mail:[email protected]

Reviewer 2 Report

This review details the applications of Herb polysaccharides based drug delivery systems for Immunomodulation. The authors has discussed different drug delivery systems in detail. A few additions/modifications could further improve the quality of the manuscript. 

1. The quality of figures are poor and can be improved. In particular, a figure detailing the concepts discussed in this review should be included

2. Authors need to write about the effect of charge, molecular weight and the degree of hydrophobicity of the HPS on immunomodulation.

3. A section discussing the pathways and receptors that HPS can target the immune cells should be included.

Author Response

Ms. Ref. No.:pharmaceutics-1803570

Title:Herb Polysaccharide-based Drug Delivery System: Fabrication, Properties, and Applications for Immunotherapy

Dear reviewer 2:

Thank you for your letter and the kind comments concerning our manuscript. We have studied comments carefully and have made appropriate and acceptable revisions. The revised parts were marked in red in the manuscript. The main corrections in the paper and the responds to the comments are as following:

This review details the applications of Herb polysaccharides based drug delivery systems for Immunomodulation. The authors has discussed different drug delivery systems in detail. A few additions/modifications could further improve the quality of the manuscript. 

  1. The quality of figures are poor and can be improved. In particular, a figure detailing the concepts discussed in this review should be included

Response: The author appreciated for the kind suggestions. We have re-drawn all the figures and made a graphic abstract (as seen in page 2, line 52.) to illustrated the concepts of the review.

  1. Authors need to write about the effect of charge, molecular weight and the degree of hydrophobicity of the HPS on immunomodulation.

Response: we have discussed the charge, molecular weight and the degree of hydrophobicity of the HPS on immunomodulation, as seen in page 4, lines 156-179.

  1. A section discussing the pathways and receptors that HPS can target the immune cells should be included.

Response: The authors quite agree with the reviewer. Accordingly, we have made a discussion on the he pathways and receptors that HPS can target the immune cells should be included, as seen in pages 3-4, lines 119-155.

In addition, we have made a thorough revision to make the paper logical and specific. Firstly, the immunomodulatory activity of mentioned HPS was summarized in section 2, and the potential mechanism was discussed from the perspective of structure-activity relationship. Second, in section 3, the principle and strategy of fabrication of HSP-DDS was described. Then, the role of HPS in DDS was summarized with emphasizing the immunomodulatory effects. Based on this, we reviewed the recent research progress of classical HPS-DDS formulations such as nanoparticles, micelles, hydrogels, microneedles in section 4. Finally, we point out the main limitations of HPS-DDS and proposed the further study directions.

Overall, we thank the kind comments from the editor and the reviewers. Those comments are all valuable and very helpful for revising and improving our paper, as well as the important guiding significance to our researches. Furthermore, a language editing services was employed. We hope now the manuscript can meet the publication standards of the journal after our corrections.

Name: Chengxiao Wang

Unit: Kunming University of Science and Technology

E-mail:[email protected]

Reviewer 3 Report

Paper/ Review summarises the applications of plant sourced carbohydrates for nanoenabled applications. Please see my comments below:

1. Title needs to be changed. "Construction" is not an ideal term. Similarly other terms like manufacture, fabrication, synthesis can be employed also throughout the text. 

2. Abstract - Herb toxicity is not documented and should not be taken for granted

3. LIne 5, remove capitalisation from liposomes. 

4. Please reference: In particular, these HPSs can also act as adjuvants, immunomodulators, and bioactive macromolecular drugs that exert synergistic effects or amplify the therapeutic activity of the primary drug - lines 59-60

5. Introduction ; It is not entirely clear in introduction how herb polysaccharides differ from polysaccharides obtained from out plants that are widely used in drug delivery systems and why special attention need to be paid to them. If structurally are the same, why is the origin of relevance? and wouldnt it more appropriate to discuss extraction methods for these polysaccharides than their applications which are widely known all ready? 

6. Introduction: Similarly many properties are proposed for herb polysaccharides but no references are provided and is there a difference that needs to be highilighted? 

7. Define medicinal plant? Really any plant can be more or less classified in this category. 

8. References missing - They have complex, important, and multi-faceted biological activities, including anti-tumor, anticoagulant, antioxidant, antiviral, immunomodulatory, hypolipidemic, and antihepatotoxic activities. Lines 77-78. References are needed for all of these properties and this platform statements are overarching with little evidence behind them. 

9. promote the growth of central immune organs - not clear what it means. 

10. Polysaccharides generally contain multiple fractions with different molecular masses, each with their own level of immune activity. - Vague. Be specific on which polysaccharides? which molecular weights? which fraction?

11. Table 1 is useful; correct antioxidation to antioxidant. 

12. Table 2 needs to be ammended and provide a lot more detail. The composition of carbohydrates particles is needed as w/w , the method of particle formation, the drug loading, particle size, polydispersity and zeta potential, and in vitro model and effects and in vivo models and effects along with any toxicological effects. The molecular weight of carbohydrates also should feature for comparison. 

13. Little information is given regarding purity and structure of carbohydrates extracted and used in described applications. 

14. Discuss regulatory likelihood of approval for these carbohydrates and and toxicological profile. 

Author Response

Ms. Ref. No.:pharmaceutics-1803570

Title: Herb Polysaccharide-based Drug Delivery System: Fabrication, Properties, and Applications for ImmunotherapyDear reviewer 3: Thanks for your letter and the kind comments concerning our manuscript. We have studied comments carefully and have made appropriate and acceptable revisions. The revised parts were marked in red in the manuscript. The main corrections in the paper and the responds to the comments are as following: 1. Title needs to be changed. "Construction" is not an ideal term. Similarly other terms like manufacture, fabrication, synthesis can be employed also throughout the text. Response: The authors quite agree with the reviewer. Accordingly, We change the title to“Herb Polysaccharide-based Drug Delivery System: Fabrication, Properties, and Applications for Immunotherapy ”, as seen in page1, lines 2-3.2. Abstract - Herb toxicity is not documented and should not be taken for grantedResponse: The authors quite agree with the reviewer. Accordingly, we have revised the abstracts and the relevant descriptions in the article,  as seen in page1, lines 13-14.3. LIne 5, remove capitalisation from liposomes.

Response:We have revised at the appropriate position.

4. Please reference: In particular, these HPSs can also act as adjuvants, immunomodulators, and bioactive macromolecular drugs that exert synergistic effects or amplify the therapeutic activity of the primary drug - lines 59-60Response: We have revised and inserted references at the appropriate position, as seen in pages 2- 3, lines 75-81.5. Introduction ; It is not entirely clear in introduction how herb polysaccharides differ from polysaccharides obtained from out plants that are widely used in drug delivery systems and why special attention need to be paid to them. If structurally are the same, why is the origin of relevance? and wouldnt it more appropriate to discuss extraction methods for these polysaccharides than their applications which are widely known all ready?

Response: As the reviewer mentioned, some polysaccharides obtained from plants have been used in DDS, such asl cellulose and alginate. Compared to these plant PS, HPS is highly featured by their strong immunomodulatory activity. Therefore, In addition to good biodegradability, HPS-based DDS also have advantages of immunotherapy. Therefore, this review focuses on the recent research progress of HPS-DDS in combination with its immunomodulatory activity and biomaterial properties. According to the comments, we have clarified the reasons for choosing the purpose of this review in the introduction section  (page3, lines 54-117).

6. Introduction: Similarly many properties are proposed for herb polysaccharides but no references are provided and is there a difference that needs to be highilighted? Response: We have revised and inserted reference at the appropriate position, as seen in pages 2-3, lines 75-106.7. Define medicinal plant? Really any plant can be more or less classified in this category. Response: The authors thank this kind comments. In our opinion, the definition of medicinal plant includes two aspects. In a broad sense, any plant with pharmacological activity can be more or less classified in this category. In a narrow sense, the medicinal plant are the ones that have been recorded in the pharmacopoeia and have been commercially available in pharmaceutical field. In this article, we use the narrow definition of medicinal plant, namely herb medicine. Accordingly, we have replaced “medicinal plant” with “herb plant”, as seen in page2, lines 76-77 , page3, lines 81-97 and 119.8. References missing - They have complex, important, and multi-faceted biological activities, including anti-tumor, anticoagulant, antioxidant, antiviral, immunomodulatory, hypolipidemic, and antihepatotoxic activities. Lines 77-78. References are needed for all of these properties and this platform statements are overarching with little evidence behind them. Response: We have revised and inserted references at the appropriate position, as seen in page 3, lines 79-81. 9. promote the growth of central immune organs - not clear what it means.  Response: We have revised into “promote the development of immune organs”, as seen in page4, lines 129-132.10. Polysaccharides generally contain multiple fractions with different molecular masses, each with their own level of immune activity. - Vague. Be specific on which polysaccharides? which molecular weights? which fraction?Response: The authors quite agree with the reviewer. Accordingly,This problem has discussed in section 2,  as seen in page 4, lines 158-167.11. Table 1 is useful; correct antioxidation to antioxidant. Response: We have made changes to replace the words in Table 1, accordingly, as seen in page12, line 402.12. Table 2 needs to be ammended and provide a lot more detail. The composition of carbohydrates particles is needed as w/w , the method of particle formation, the drug loading, particle size, polydispersity and zeta potential, and in vitro model and effects and in vivo models and effects along with any toxicological effects. The molecular weight of carbohydrates also should feature for comparison.

Response: According to your suggestion, we have sorted out curing disease, the preparation methods and the role of HPS in DD in Table 2, as seen in page 12, line 402. In order to avoid the length of the article, there is not discussedt that the particle size, polydispersity , zeta potential extraction method, and other effects of HPS on immune activity in detail in each chapter. Therefore, we do not showed these metrics in Table 2.

 13. Little information is given regarding purity and structure of carbohydrates extracted and used in described applications.

Response: The authors thank for the kind suggestion. However, the information of HSP purity is lacked in the major proportions of the references. In our opinion, the bioactivities of PS are closely related with their purity. For example, trace amounts of lipopolysaccharide in PS would significantly affect their immunoregulatory activities. In our previous work, the purities of HSP was strictly controlled (>95%) for fabrication of HSP-DDS . Accordingly, we have made a discussion about the purity in the article, as seen in section 5,  as seen in pages 23-24, lines 745-758.

14.Discuss regulatory likelihood of approval for these carbohydrates and and toxicological profile. 

Response: The authors quite agree with the reviewer. Accordingly, we added the regulatory likelihood of approval for HPS in the discussion sections, as seen in page 23, lines 730-744.In addition, we have made a thorough revision to make the paper logical and specific. Firstly, the immunomodulatory activity of mentioned HPS was summarized in section 2, and the potential mechanism was discussed from the perspective of structure-activity relationship. Second, in section 3, the principle and strategy of fabrication of HSP-DDS was described. Then, the role of HPS in DDS was summarized with emphasizing the immunomodulatory effects. Based on this, we reviewed the recent research progress of classical HPS-DDS formulations such as nanoparticles, micelles, hydrogels, microneedles in section 4. Finally, we point out the main limitations of HPS-DDS and proposed the further study directions.

Overall, we thank the kind comments. Those comments are all valuable and very helpful for revising and improving our paper, as well as the important guiding significance to our researches. Furthermore, a language editing services was employed. We hope now the manuscript can meet the publication standards of the journal after our corrections.

Name: Chengxiao Wang

Unit: Kunming University of Science and Technology

E-mail:[email protected]

Round 2

Reviewer 1 Report

Authors did good job in improving their manuscript and generally it can be accepted but there are some new remarks which were detected in modified version

Line 82 "Chinese medicine that invigorates qi, spleen, and lung has long" - please explain what does it mean?

Table 2 “Inulin polysaccha-rides”, please correct this since inulin is the polysaccharide itself.

Author Response

Ms. Ref. No.:pharmaceutics-1803570

Title: Herb Polysaccharide-based Drug Delivery System: Fabrication, Properties, and Applications for Immunotherapy

Dear Reviewer #1:

Thank you for your comments concerning our manuscript again. Those comments are all valuable and very helpful for revising and improving our paper. We have studied comments carefully and have made correction by “track changes” functions of MS word.

  1. Line 82 "Chinese medicine that invigorates qi, spleen, and lung has long" - please explain what does it mean?

Response: The authors apologized for puzzling the reviewer. The words such as “Qi’, “Yin” and “Yang” are specific terms from ancient Chinese medicine. In order to ensure the consistency of the words used throughout the full text, we have deleted these descriptions, which would reduce the reading experience (page 3, lines 75-93).

2.Table 2 “Inulin polysaccha-rides”, please correct this since inulin is the polysaccharide itself.

Response: We are very sorry for our negligence. Accordingly, we have revised the error in Table 2, as seen in page12, line 398.

In addition, special thanks to you for your good comments. We hope now the manuscript can meet the publication standards of the journal after our corrections.

Chengxiao Wang

Kunming University of Science and Technology

E-mail:[email protected]

Reviewer 3 Report

The issue of purity and characterisation of the herb carbohydrates is not addressed in detail and Table 2 would require a lot more detail to be of value to reader as this is a review.

Author Response

Ms. Ref. No.:pharmaceutics-1803570

Title: Herb Polysaccharide-based Drug Delivery System: Fabrication, Properties, and Applications for Immunotherapy

Dear Reviewer #3:

Thank you for your comments concerning our manuscript again. We have studied comments carefully and have made correction by “track changes” functions of MS word. 

1.The issue of purity and characterisation of the herb carbohydrates is not addressed in detail

Response: The authors appreciated for the constructive suggestions of the reviewer. We have consulted a number of relevant articles but failed to obtain the specific information about the purity of HSP used in DDS.  In the reported works, the HSP used in DDS are first purified by DEAE column to remove the general impurities (e.g. oligosaccharides, proteins and lignin). Then, the HSP are further purified by sephadex column including G20, G50, G100 and G200 to obtain the homogeneous products with narrow Mw distribution. As a result, the HSP produced by DEAE and sephadex column purification usually have an acceptable purity above 90%. In our previous work, PNPS, MP and BSP were used to fabricate the DDS such as micelle, microneedles and hydrogel. the purities for all there HSP were all above 95%, Accordingly, the purity of HSP in DDS was discussed , as seen in pages 24-25, lines739-750.2.Table 2 would require a lot more detail to be of value to reader as this is a review.Response: Table 2 have been further enriched with particle size, zeta potential of DDS and target cells, as seen in page12, line 398.

In addition, special thanks to you for your good comments. We hope now the manuscript can meet the publication standards of the journal after our corrections.

Chengxiao Wang

Kunming University of Science and Technology

E-mail:[email protected]
